

# Accounting for effects of coagulation and model uncertainties in particle number concentration estimates based on measurements from sampling lines – A Bayesian inversion approach with SLIC v1.0

Matti Niskanen[1], Aku Seppänen[1], Henri Oikarinen[1], Miska Olin[2], Panu Karjalainen[3], Santtu Mikkonen[1,4], and Kari Lehtinen[1]

[1]Department of Technical Physics, University of Eastern Finland, Kuopio, Finland
[2]Department of Atmospheric Sciences, Texas A&M University, College Station, TX, United States
[3]Aerosol Physics Laboratory, Tampere University, Tampere, Finland
[4]Department of Environmental and Biological Sciences, University of Eastern Finland, Kuopio, Finland
**Correspondence:** Matti Niskanen (matti.niskanen@uef.fi)

**Abstract.** The particle number (PN) emissions of both light- and heavy-duty vehicles are nowadays regulated, and are typically measured from a full dilution tunnel with constant volume sampling (CVS). PN measurements for research and development purposes, though, are often taken from the raw exhaust to avoid the high set up costs of CVS. There is, however, a risk with these and any other kind of PN measurements with high number concentrations, that physical processes such as coagulation and

diffusion losses inside sampling lines can alter, sometimes dramatically, the particle size distribution and bias its measurement. In this paper, we propose a method in the Bayesian framework for inverse problems to estimate the initial, unaltered, particle size distribution, based on the distorted measurements. The proposed method takes into account particle morphology and van der Waals/viscous forces in the coagulation model, allows the incorporation of prior information on the particle size distribution and, most importantly, a systematic quantification of uncertainty. We analyze raw exhaust PN measurements of a

fuel-operated auxiliary heater, and find that while a typical sampling line can reduce the PN by more than 50 %, the initial particle size distribution can be feasibly estimated with reasonable computational demands. The proposed method should give more freedom for designing the measurement set up and also aid in the comparison of results obtained at different sampling locations, such as CVS and tailpipe.

## 1 Introduction

Particulate matter (PM), i.e., particle pollution, poses great health risks to humans and causes environmental damage (Shiraiwa et al., 2017; Manisalidis et al., 2020). Significant PM sources, such as vehicle tailpipe emissions, are therefore in most cases subject to particle mass regulations, and nowadays also to particle number (PN) regulations in Europe (Commission Regulation (EU) 2019/1939; Giechaskiel et al., 2021b). PN regulations were introduced to better cover the fraction of ultrafine ($< 0.1\ \mu$m) particles, which is negligible in terms of mass but dominates the ambient atmosphere in terms of PN (80-90 % of all particles)



(Hofman et al., 2016). These ultrafine particles are causing a growing concern in the public health community due to their ability to circumvent primary airway defenses and penetrate deep in the lungs (Kwon et al., 2020; Morawska and Buonanno, 2021). Furthermore, the harmful systematic health effects associated with coarse ($< 10$ $\mu$m) and fine ($< 2.5$ $\mu$m) particulates can often be attributed to the fraction of ultrafine particles (Kwon et al., 2020). At the moment PN regulations are given for particles with diameter $> 23$ nm, not because smaller particles are any less dangerous, but mainly because smaller particles

are very sensitive to sampling conditions and thus hard to measure accurately and with good repeatability (Giechaskiel et al., 2021b). The Euro 7 road vehicles emission standard proposal increases the size range to particles $> 10$ nm (Giechaskiel et al., 2024). Details of the emissions such as those emitted from tailpipes are also important basic knowledge for the purpose of climate change research, as aerosol forcing uncertainty represents the largest climate forcing uncertainty overall (Kahn et al., 2023).

PN emissions are typically measured using condensation particle counters (e.g., Hering et al. (2005)) or diffusion chargers (e.g., Schriefl et al. (2019)), which can measure total PNs. If more detailed information is required, one can measure PN as a function of particle size, i.e., a particle size distribution (PSD), with an instrument such as the Engine Exhaust Particle Sizer (EEPS) spectrometer (TSI.; Wang et al., 2016a). There are, however, aspects of PN measurement that may bias the results considerably unless explicitly accounted for in the design of the experiment (which may not always be possible), or numerically

post-measurement. For example, measuring PSD in raw exhaust usually involves the use of sampling lines of different lengths to transfer and possibly cool the sample to the measurement devices. Within the sampling lines, the particles have a window of time to coagulate and diffuse onto the inner walls of the lines, which alters the PSD of the raw exhaust. Shortest possible residence times in the lines are typically targeted, because short enough residence times (and low enough PN concentrations) limit the effects of diffusion and coagulation, leaving the PSD mostly unaltered. This may not always be practicable, however,

and sometimes the PSD undergoes significant changes, up to more than an order of magnitude, especially in the concentrations of the smallest particle sizes. In one study, Giechaskiel et al. (2019) examined the measurement of PN directly from the tailpipe of heavy-duty engines and found that an increase from 0.5 meters to 4 meters in the sampling line length resulted in a loss of 20-50 % for particles $> 10$ nm, not explainable with just diffusion losses. Further, Liu et al. (2021) studied the effect of Brownian coagulation on particle evolution in gasoline engine exhaust by measuring the PSD at various locations along a

laboratory exhaust system. They found that the PSD evolution mainly occurs for sizes $< 50$ nm, the evolution is dominated by coagulation, and that a significant, up to 90 %, particle reduction can occur when there is a high number of accumulation mode ($> 50$ nm) particles. Such high losses would render a measurement useless unless the changes in the PSD are rigorously quantified and corrected for.

Of the physical processes in a sampling line, wall deposition losses are nowadays commonly corrected, but the effect of

coagulation is typically not. Further, because the effect of coagulation is nonlinear and depends on the PN concentration it cannot be corrected with a simple calibration measurement. To ensure coagulation and diffusion don't have a significant effect on the PSD, legislation specifies for example the maximum residence time and Reynolds number for a sample in the sampling line (Giechaskiel et al., 2021b), but we haven't found in the literature attempts to correct the estimate of a PSD after the measurement has been made. In this paper, we aim to find out if the initial PSD can be estimated based on measurements that



have been distorted by coagulation and diffusional losses. Our approach uses methods in the Bayesian framework for inverse problems (Kaipio and Somersalo, 2006), and makes use of a numerical model of the coagulation and wall diffusion process that takes into account the fractal nature of soot particles (Park et al., 2003; Rogak and Flagan, 1992) as well as van der Waals and viscous forces (Alam, 1987). Due to uncertainties related to the measurement (e.g., noise) and to the coagulation model (e.g., poorly known parameters), the resulting estimate is also uncertain to some degree. In the Bayesian framework, this uncertainty is quantified by the posterior probability density, i.e., the probability density of the unknown variable of interest, conditioned on the measurement. The possible prior information on the model unknown as well as statistics of the measurement noise are also accounted for. From the posterior one can calculate not only the most probable values of the initial PSD, but also the credible intervals to explicitly quantify the uncertainty of the PSD estimate. Robust uncertainty quantification is essential to assess the reproducibility of measurements between different laboratories (Giechaskiel et al., 2018, 2021a).

To test the proposed method, we carry out inversion with both synthetic data and real PSD measurements of a fuel-operated auxiliary heater (Oikarinen et al., 2022). We will compute the estimate of the initial PSD in two ways that are useful in different circumstances. The first one is based on solving an optimization problem, which is computationally fast and can be done in real time when collecting measurements in the field. In this case, the posterior uncertainty is approximated with a Gaussian distribution. The second approach is based on sampling the posterior with Markov chain Monte Carlo (MCMC), which is a lot more demanding computationally, but doesn't need to rely on Gaussian approximations for the uncertainty estimates. The second approach can also be used to take into account the possible uncertainty in the parameters of the coagulation model, as will be shown later. We will also compare and discuss the results between the two approaches.

The remainder of the paper is organized as follows. In Sect. 2, we describe our approach to modelling coagulation and wall deposition. In Sect. 3, we describe and solve the inverse problem of estimating the initial PSD based on measurements done with a system that has a sampling line. In Sect. 4, we apply the method on synthetic and real data, and the results are discussed in Sect. 5. Finally, conclusions are given in Sect. 6.

## 2   Simulating the microphysical processes in a sampling line

When collecting PSD data from sampling lines, there are two main processes quick enough to affect aerosol particles during the short time (up to a few seconds) a sample resides in a sampling line: coagulation (agglomeration of particles), and wall deposition due to diffusion (particles lost to the sampling line walls). Together these processes reduce the number of particles and increase their size.

The time evolution of a PN concentration density $n(v,t)$ (unit: $\mathrm{m^{-1}cm^{-3}}$) experiencing coagulation and wall diffusion can be described by the following equation (Seinfeld and Pandis, 2016)

$$\frac{dn(v,t)}{dt} = \frac{1}{2}\int_0^v \beta(v-\bar{v},\bar{v})n(v-\bar{v},t)n(\bar{v},t)\,\mathrm{d}\bar{v} - n(v,t)\int_0^\infty \beta(\bar{v},v)n(\bar{v},t)\,\mathrm{d}\bar{v} - R(v)n(v,t),\tag{1}$$





where the first integral represents a coagulation source, the second a coagulation sink, and $R(v)$ in this case the rate of removal of particles by wall diffusion. Particle volume and time are denoted by $v$ and $t$, respectively. The rate of coagulation is governed by the coagulation coefficient $\beta(v_1, v_2)$, which describes the frequency two particles of volume $v_1$ and $v_2$ collide.

To approximate the solution of Eq. (1) numerically, the particle size range is first discretized. We adopt here the sectional method (Gelbard et al., 1980; Jacobson et al., 1994; Lehtinen and Zachariah, 2001) for its computational efficiency and ease of implementation, where the particle size range is divided into a finite number of sections, or bins. Here we use geometrically distributed bins so that the logarithm of the bin width is constant. The sectional method conserves total particle volume, and in Salminen et al. (2022) it was even found that for pure coagulation the sectional method can be more accurate than the finite element method with a similar number of discretization points.

In a typical implementation of the sectional method, particles within each bin are approximated to have a constant size equal to the bin middle point. Consider then two particles from bins with volumes $v_i$ and $v_j$ colliding, and forming a larger particle with volume $v_i + v_j$. Unless the volume of the new particle coincides exactly with a center of a bin, the particle is divided into two adjacent bins, in a way that conserves the total volume. For this purpose, let us define a size-splitting operator $\xi_{ijk}$, which gives the volume fraction of the coagulated particle $v_i + v_j$ partitioned into bin $k$:

$$
\xi_{ijk} = \begin{cases}
\dfrac{v_{k+1} - (v_i + v_j)}{v_{k+1} - v_k}, & v_k \leq v_i + v_j < v_{k+1}; & k < n_B \\
1 - \xi_{ij(k-1)}, & v_{k-1} < v_i + v_j < v_k; & k > 1 \\
1, & v_k \leq v_i + v_j; & k = n_B \\
0, & \text{otherwise},
\end{cases}
\tag{2}
$$

where $n_B$ is the total number of bins. Let us now denote the PN concentration in the $k$-th bin by $N_k = n_k \Delta D_p$, where $\Delta D_p$ is the width, and $n_k$ the PN concentration density (cf. $n(v, t)$ in Eq. (1)), of bin $k$, respectively, and $k = 1, \ldots, n_B$. The discretised version of Eq. (1) then reads

$$
\frac{dN_k}{dt} = \frac{1}{2} \sum_{i,j} \xi_{ijk} \beta_{ij} N_i N_j - \sum_i \beta_{ik} N_i N_k - R_k N_k,
\tag{3}
$$

where $\beta_{ij}$ is the coagulation coefficient between size bins $i$ and $j$. Time integration for Eq. (3) can be carried out for example with Runge-Kutta methods. We use here the RK45 as implemented in SciPy[1].

## 2.1 Coagulation model

Particles collide due to their Brownian, or thermal, motion, and motion due to external forces such as laminar shear or turbulent flow, gravitational, van der Waals, Coulomb, and hydrodynamic forces (Seinfeld and Pandis, 2016). Colliding liquid particles coalesce and retain their spherical shape. Solid particles above a certain critical size $r_1$, however, are not expected to coalesce but instead form fractal-like agglomerates of dense primary particles. The collision frequency for fractal-like particles can be significantly higher than for spherical particles with the same volume (Rogak and Flagan, 1992).

---

[1]A package for scientific computing algorithms in Python.



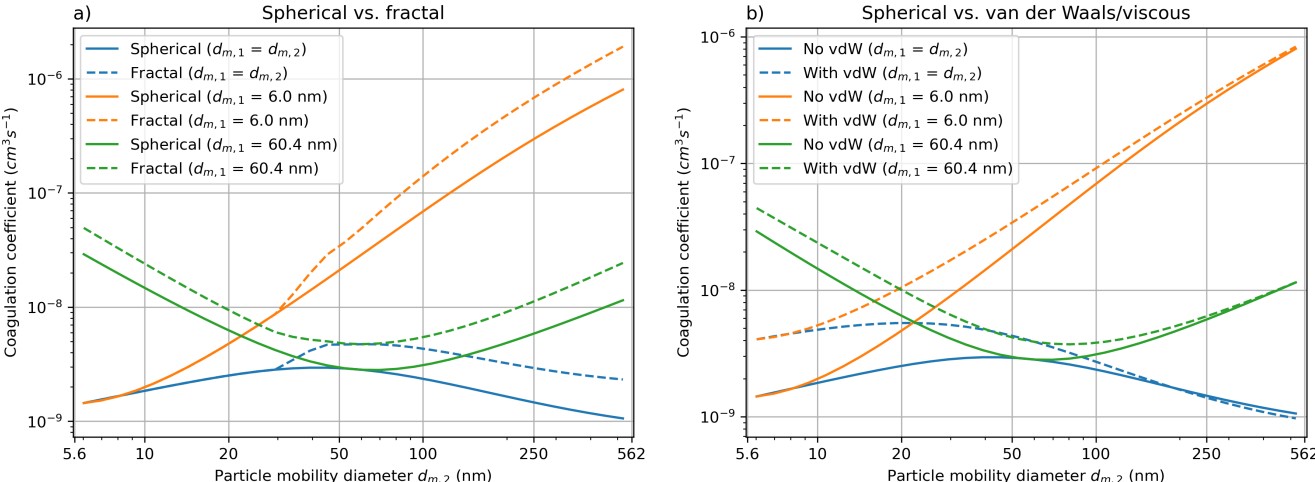

**Figure 1.** Coagulation coefficients calculated for mobility diameters $5.62 - 562$ nm. a) Spherical vs. fractal coagulation coefficient (fractal dimension $= 1.7$ and primary particle diameter $= 30$ nm). b) Spherical coagulation coefficient with and without van der Waals and viscous forces (Hamaker constant $= 2 \cdot 10^{-19}$ J).

In this work, for coagulation we consider the particles' Brownian motion and the effects of van der Waals and viscous forces, as those are likely to dominate over other factors in coagulation at short time scales and distances. Further, in our specific example with soot particles, we will take into account the particle shape because soot particles are known to form 115 long chains of aerosol agglomerates (Park et al., 2003). Figure 1 demonstrates the effect of fractal geometry and van der Waals/viscous forces on the coagulation coefficient.

### 2.1.1 Brownian coagulation of fractal-like agglomerates

In this paper, coagulation of fractal-like agglomerates is modelled following mostly Rogak and Flagan (1992) and Jacobson and Seinfeld (2004), where it is assumed that the agglomerate structure is completely determined by three parameters: the 120 fractal dimension $D_{\mathrm{f}}$, the primary particle radius $r_1$, and the number of primary particles $\mathcal{N}$. All primary particles are thus assumed to be of the same size. The fractal dimension, $D_{\mathrm{f}} \in [1,3]$, characterizes the shape of the agglomerate. For example, when $D_{\mathrm{f}} = 1$ the agglomerate forms a line whereas with $D_{\mathrm{f}} = 2$ it fills space like a surface, and as $D_{\mathrm{f}}$ approaches 3, it curls up to resemble a spherical volume. For soot particles, typical values for the primary particle radius are between 10 to 17 nm and the fractal dimension between 1.5 to 2.2 (Wentzel et al., 2003; Lapuerta et al., 2017).

The three parameters above give rise to a number of equivalent radii that characterize the coagulation of non-spherical particles. Let us first highlight two here: the *volume-equivalent* and the *mobility radius*. The volume-equivalent radius, $r_v$, is the radius of a sphere with the same volume as the aggregate, whereas the mobility radius, $r_{\mathrm{m}}$, is the radius of a sphere that experiences the same drag force as the agglomerate under the same dynamic conditions (Rogak and Flagan, 1992). This distinction is important, because while mobility radii are what is usually measured, the size-splitting coefficient (2) should be





calculated using the volume-equivalent radii to conserve particle volume. Thus, for non-spherical particles, the output bin sizes of a typical measurement device should not be used directly for modelling coagulation.

Let us first state $\mathcal{N}_i$ as the ratio of the volumes of the aggregate and the primary particle, $\mathcal{N}_i = (r_{v,i}/r_1)^3$, where $i$ refers to the index of a size bin. The *fractal* or *outer radius* of the agglomerate is then defined as

$$r_{\mathrm{f},i} = \begin{cases} r_1 \mathcal{N}_i^{1/D_{\mathrm{f}}}, & r_{v,i} \geq r_1, \\ r_{v,i}, & r_{v,i} < r_1. \end{cases} \tag{4}$$

The Brownian coagulation rate between two particles of sizes $i$ and $j$ can be stated in the Fuchs form, modified for fractal geometry, as

$$\beta_{i,j} = 4\pi(r_{\mathrm{c},i} + r_{\mathrm{c},j})(D_{\mathrm{m},i} + D_{\mathrm{m},j})C_{i,j}^{-1}, \tag{5}$$

where $r_{\mathrm{c},i}$ and $D_{\mathrm{m},i}$ are the collision radius and Brownian diffusion coefficient of particle $i$, respectively, and $C_{i,j}$ is a transition-regime correction factor that extends the validity of the formula from the continuum regime to sizes smaller or comparable to the mean free path of diffusion particles. Following Rogak and Flagan (1992), we set the collision radius equal to the outer radius of the particle, $r_{\mathrm{c},i} = r_{\mathrm{f},i}$.

The expressions for $D_{m,i}$ and $C_{i,j}$ require the (transition-regime) mobility radius $r_{m,i}$, which is computed by interpolating between the continuum-regime mobility radius $r_{mc,i}$ and free-molecular regime mobility radius $r_{mk,i}$, as

$$\frac{r_{\mathrm{m},i}}{C_c(\mathrm{Kn}_{\mathrm{m},i})} = \frac{r_{\mathrm{mc},i}}{C_c(\mathrm{Kn}_{\mathrm{a},i})}, \tag{6}$$

where $C_c(\mathrm{Kn})$ is the Cunningham slip correction factor, given by

$$C_c(\mathrm{Kn}) = 1 + \mathrm{Kn}(1.257 + 0.4\exp(-1.1/\mathrm{Kn})), \tag{7}$$

with Kn representing the Knudsen number. The mobility radius Knudsen number is given by

$$\mathrm{Kn}_{\mathrm{m},i} = \frac{\lambda_a}{r_{\mathrm{m},i}}, \tag{8}$$

and the adjusted Knudsen number by

$$\mathrm{Kn}_{\mathrm{a},i} = \frac{\lambda_a r_{\mathrm{mc},i}}{r_{\mathrm{mk},i}^2}. \tag{9}$$

Above, $\lambda_a$ denotes the mean free path of an air molecule, which can be calculated from

$$\lambda_a = \frac{k_{\mathrm{B}}T}{\sqrt{2}\pi d^2 p}, \tag{10}$$

where $k_{\mathrm{B}}$ is the Boltzmann constant, $T$ is the gas temperature, $d$ is the diameter of an air molecule (approximated here as the diameter of a nitrogen molecule), and $p$ is the gas pressure. The free-molecular regime mobility radius, $r_{\mathrm{mk},i}$, is assumed to be equal to the projected-area equivalent radius $r_{\mathrm{A},i}$, given by

$$r_{\mathrm{A},i} = \begin{cases} r_1 \sqrt{\max\left\{N_i^{2/3}, \min\left[1 + 0.67(N_i - 1), D_{\mathrm{f}}N_i^{2/D_{\mathrm{f}}}/3\right]\right\}}, & r_{v,i} \geq r_1, \\ r_{v,i}, & r_{v,i} < r_1, \end{cases} \tag{11}$$





and the continuum-regime mobility radius is given by

$$r_{\mathrm{mc},i} = \max\left\{ r_{\mathrm{f},i}\left[\ln\left(2\frac{r_{\mathrm{f},i}}{r_1}\right)+1\right]^{-1}, r_{\mathrm{f},i}\left(\frac{D_{\mathrm{f}}-1}{2}\right)^{0.7}, r_{\mathrm{A},i}\right\}. \tag{12}$$

Now, the diffusion coefficient can be calculated from

160 $$D_{\mathrm{m},i} = \frac{k_{\mathrm{B}}T}{6\pi\eta r_{\mathrm{m},i}}C_c(\mathrm{Kn}_{\mathrm{m},i}), \tag{13}$$

where $\eta$ represents the dynamic viscosity of gas, which is temperature-dependent and over the range of 100-1800 K can be determined using the Sutherland equation (Hinds, 1999):

$$\eta = \frac{1.458\cdot10^{-6}\,\mathrm{Pa\cdot s\cdot K}\left(\dfrac{T}{\mathrm{K}}\right)^{1.5}}{T+110.4\,\mathrm{K}}. \tag{14}$$

The correction factor $C_{i,j}$ is given by

165 $$C_{i,j} = \frac{r_{\mathrm{c},i}+r_{\mathrm{c},j}}{r_{\mathrm{c},i}+r_{\mathrm{c},j}+\sqrt{\delta_{\mathrm{m},i}^2+\delta_{\mathrm{m},j}^2}} + \tilde{c}, \tag{15}$$

where

$$\delta_{\mathrm{m},i} = \frac{(2r_{\mathrm{m},i}+\lambda_{\mathrm{m},i})^3-(4r_{\mathrm{m},i}^2+\lambda_{\mathrm{m},i}^2)^{3/2}}{6r_{\mathrm{m},i}\lambda_{\mathrm{m},i}} - 2r_{\mathrm{m},i} \tag{16}$$

and

$$\tilde{c} = \frac{4(D_{\mathrm{m,i}}+D_{\mathrm{m,j}})}{\sqrt{\bar{v}_i^2+\bar{v}_j^2}(r_{\mathrm{c},i}+r_{\mathrm{c},j})}. \tag{17}$$

The effective mean free path of a particle size $i$ is given by

$$\lambda_{\mathrm{m},i} = \frac{8D_{\mathrm{m},i}}{\pi\bar{v}_i}, \tag{18}$$

with the mean thermal speed

$$\bar{v}_i = \sqrt{\frac{8k_{\mathrm{B}}T}{\pi\bar{M}_i}}, \tag{19}$$

where $\bar{M}_i = 4/3\pi r_{v,i}^3\rho$ is the mass of a single particle, with $\rho$ denoting the particle density.

The above model was written in terms of the volume-equivalent radius $r_v$ and parameters $D_{\mathrm{f}}$ and $r_1$ to find the mobility radius $r_m$, i.e. it forms a relation $f: r_v \mapsto r_m$. On the other hand, measurements of PN concentration are usually modelled with respect to $r_m$, and hence, a mapping from $r_m$ to $r_v$ is needed. An explicit solution $f^{-1}: r_m \mapsto r_v$ is not possible, though, and we calculate $r_v$ by solving a minimization problem:

$$r_{v,i} = \operatorname*{argmin}_{r_{v,i}}\|r_{m,i}-f(r_{v,i},r_1,D_{\mathrm{f}})\|_2^2. \tag{20}$$





### 2.1.2 Van der Waals and viscous forces

Van der Waals forces are weak fluctuating dipole-dipole forces that increase the rate of aerosol coagulation (Alam, 1987). Viscous forces, on the other hand, slow down the rate of coagulation. Viscous forces are absent in the free molecular regime, but in the continuum regime can even slow down coagulation more than van der Waals forces enhance it (Jacobson and Seinfeld, 2004). These forces can be taken into account by multiplying the Brownian coagulation coefficient (5) by a correction factor $V_{i,j}$. We follow here Alam (1987), who determined an interpolation formula for the correction factor between the free-molecular and continuum regimes as

$$V_{i,j} = \frac{W_{c,i,j}\,[1+\tilde{c}]}{1 + W_{c,i,j}/W_{k,i,j}\,\tilde{c}}, \tag{21}$$

where $\tilde{c}$ is defined in (17) and $W_{k,i,j}$ and $W_{c,i,j}$ are correction factors for the free-molecular and continuum regimes, respectively. These are given by

$$W_{k,i,j} = \frac{-1}{2(r_i+r_j)^2 k_B T} \int\limits_{r_i+r_j}^{\infty} \left( \frac{dE_{i,j}(r)}{dr} + r\frac{d^2 E_{i,j}(r)}{dr^2} \right) \exp\left[ \frac{-1}{k_B T}\left( \frac{r}{2}\frac{dE_{i,j}(r)}{dr} + E_{i,j}(r) \right) \right] r^2 dr, \tag{22}$$

and

$$W_{c,i,j} = \left[ (r_i+r_j) \int\limits_{r_i+r_j}^{\infty} \frac{D^\infty}{D_{i,j}}(r) \exp\left( \frac{E_{i,j}(r)}{k_B T} \right) \frac{dr}{r^2} \right]^{-1}. \tag{23}$$

The correction factors include the Van der Waals interaction potential

$$E_{i,j}(r) = -\frac{A}{6}\left[ \frac{2r_i r_j}{r^2-(r_i+r_j)^2} + \frac{2r_i r_j}{r^2-(r_i-r_j)^2} + \ln\frac{r^2-(r_i+r_j)^2}{r^2-(r_i-r_j)^2} \right], \tag{24}$$

where $A$ is the Hamaker constant, and the diffusion ratio that corrects the diffusion coefficient in the continuum regime for viscous forces:

$$\frac{D^\infty}{D_{i,j}}(r) = 1 + \frac{2.6 r_i r_j}{(r_i+r_j)^2}\sqrt{\frac{r_i r_j}{(r_i+r_j)(r-r_i-r_j)}} + \frac{r_i r_j}{(r_i+r_j)(r-r_i-r_j)}. \tag{25}$$

Typical values for the Hamaker constant of soot are in the range of $1\times 10^{-19} - 4\times 10^{-19}$ J (Liu et al., 2018).

### 2.2 Wall deposition by diffusion

Aerosol particles adhere when they collide with a surface, and thus their concentration at the surface is zero. This results in a concentration gradient near the surface which causes a continuous diffusion of particles to the surface. For laminar flow of particles through a cylindrical tube, the fraction $P = n_{\text{out}}/n_{\text{in}}$ of entering particles that exit can be approximated with the help of a dimensionless diffusion parameter $\mu$ as (Hinds, 1999)

$$P = \begin{cases} 1 - 5.50\mu^{2/3} + 3.77\mu, & \mu < 0.009, \\ 0.819\exp(-11.5\mu) + 0.0957\exp(-70.1\mu), & \mu \geq 0.009. \end{cases} \tag{26}$$



The diffusion parameter is defined as

$$\mu = \frac{DL}{Q},\tag{27}$$

where $L$ is the tube length, and $Q$ is the volume flow rate $(\mathrm{m^3 s^{-1}})$ through the tube. For the particle diffusion coefficient $D$ we use $D_{\mathrm{m}}$ from Eq. (13). To transform the penetration $P$ into the removal term $R$ in Eq. (3), we set

$$R = -\log(P)/T,\tag{28}$$

where $T$ is the residence time of the aerosol in the tube.

## 3 Estimating the initial size distribution

Let us now formulate the problem of estimating the initial PSD as a problem of statistical inference within the Bayesian framework. The unknown parameters are modelled as random variables, and all information on them is expressed in terms of probability density functions. The solution to the inference problem is the posterior probability density, a conditional probability density of the unknown parameters conditioned on the measured data (Kaipio and Somersalo, 2006).

Let us denote by $\tilde{\mathbf{N}} \in \mathbb{R}^{n_B}$ a vector that represents a parametrization of the PSD of the aerosol, at the entry of the sampling line (i.e., the initial PSD). This will be the model unknown in the Bayesian inference. In the parametrization used below, $\tilde{\mathbf{N}}$ consists of the logarithms of the size-discretized PSD $\mathbf{N}$, $\tilde{\mathbf{N}} := \log \mathbf{N}$, to enforce positivity and to treat the wide range of number concentrations in a numerically more stable way. In what follows, $\tilde{\cdot}$ will be used to denote a variable on the log-scale. Further, let $\mathbf{y} \in \mathbb{R}^{n_M}$ be a vector consisting of noisy, indirect observations of the PSD after the particles have undergone coagulation and diffusion processes when transported through the sampling line. The posterior is then given by Bayes' formula,

$$\pi(\tilde{\mathbf{N}}|\mathbf{y}) = \frac{\pi(\mathbf{y}|\tilde{\mathbf{N}})\pi(\tilde{\mathbf{N}})}{\pi(\mathbf{y})} \propto \pi(\mathbf{y}|\tilde{\mathbf{N}})\pi(\tilde{\mathbf{N}}),\tag{29}$$

where $\pi(\mathbf{y}|\tilde{\mathbf{N}})$ is the likelihood function, $\pi(\tilde{\mathbf{N}})$ is the prior probability density, and $\pi(\mathbf{y})$ the value of the marginal probability density of measurements at $\mathbf{y}$, which once the measurements are realized, acts as a normalization constant. The likelihood function describes the likelihood of different measurement outcomes given a realisation of parameters $\tilde{\mathbf{N}}$. It considers statistics of the measurement noise and the misfit between measurements and the model. The prior density models statistically the information on the parameters before measurements $\mathbf{y}$ are accounted for. It should include a higher probability for parameter values we expect to see versus those we do not expect to see, and can, for example, include an assumption that small differences in number concentration between neighboring size bins are more probable than large ones.

### 3.1 The prior

We model the prior density as Gaussian with mean $\tilde{\mathbf{N}}_*$ and covariance $\tilde{\Gamma}_{\mathrm{pr}}$. The covariance is constructed to promote smoothness (Lieberman et al., 2010) between adjacent size bins:

$$\tilde{\Gamma}_{\mathrm{pr}}(i,j) = \tilde{a} \exp\left\{-\frac{1}{2}\frac{\|\tilde{d}_i - \tilde{d}_j\|^2}{\tilde{b}^2}\right\},\tag{30}$$



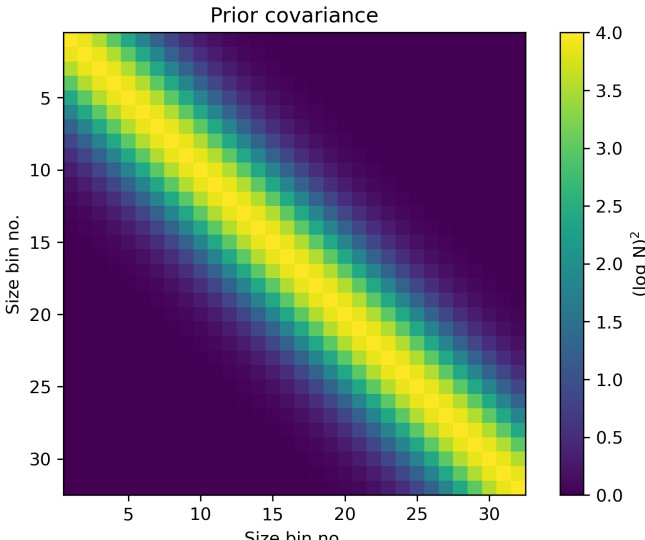

**Figure 2.** An example prior covariance matrix with size bins covering two orders of magnitude and 16 bins per decade.

where $\tilde{a} = \tilde{\Gamma}_{\mathrm{pr}}(i, i)$ is the variance of the initial PSD, $\tilde{d}_i$ are the size bin centers, and $\tilde{b} = \tilde{l}/\sqrt{2\ln(100)}$ is related to the smoothness of the initial PSD over the size range. The degree of smoothness is controlled by correlation length $\tilde{l}$, which is defined as the distance (here, in terms of particle size on a logarithmic scale) over which the cross-covariance $\tilde{\Gamma}_{\mathrm{pr}}(i, j)$ drops to 1 % of $\tilde{a}$. Due to the $\log$-transformation, a correlation length $\tilde{l} = 1$ corresponds to one order of magnitude. An example of a covariance matrix with $\tilde{l} = 3/4$ and $\tilde{a} = 4$, when each order of magnitude is divided into 16 bins, is shown in Fig. 2.

With the above notation, the prior density can be written as

$$\pi(\tilde{\mathbf{N}}) \propto \exp\left\{ -\frac{1}{2}\|\tilde{L}_{\mathrm{pr}}\left(\tilde{\mathbf{N}}_* - \tilde{\mathbf{N}}\right)\|^2 \right\}, \tag{31}$$

where $\tilde{L}_{\mathrm{pr}}$ is a matrix square root of the inverse of the prior covariance, i.e. $\tilde{L}_{\mathrm{pr}}^T \tilde{L}_{\mathrm{pr}} = \tilde{\Gamma}_{\mathrm{pr}}^{-1}$.

### 3.2 The likelihood

Let $h : \mathbb{R}^{n_B} \to \mathbb{R}^{n_M}$ denote the parameter-to-observable mapping, i.e., the *forward model*, which models the dependency between the initial time PSD (or, to be specific, its logarithmic parameters) and the observable quantity. That is, in this case, *h not only* models the measurement device but *also* the coagulation and wall deposition processes that the particle ensemble undergoes while being transported through the sampling line between the inital time and the time of the observation. We will derive the exact form of $h$ in Sect. 4 because it depends on the measurement device used. With the standard assumption that the measurement process is corrupted by additive Gaussian noise $\mathbf{e} \in \mathbb{R}^{n_M}$, the following observation model can be written for the measurement

$$\mathbf{y} = h(\tilde{\mathbf{N}}) + \mathbf{e}. \tag{32}$$





If we assume that the measurement noise is independent of the unknkowns, and normally distributed with zero mean and covariance $\Gamma_e$, $\mathbf{e} \sim \mathcal{N}(0, \Gamma_e)$, the likelihood can be written as

$$\pi(\mathbf{y}|\tilde{\mathbf{N}}) \propto \exp\left\{-\frac{1}{2}\left\|L_e\left(\mathbf{y} - h(\tilde{\mathbf{N}})\right)\right\|^2\right\}, \tag{33}$$

where $L_e^T L_e = \Gamma_e^{-1}$.

## 3.3 The posterior

The solution to the inference problem, the posterior density (29), can now be stated by combining the prior (31) and the likelihood (33):

$$\pi(\tilde{\mathbf{N}}|\mathbf{y}) \propto \exp\left\{-\frac{1}{2}\left\|L_e\left(\mathbf{y} - h(\tilde{\mathbf{N}})\right)\right\|^2 - \frac{1}{2}\left\|L_{\mathrm{pr}}(\tilde{\mathbf{N}}_* - \tilde{\mathbf{N}})\right\|^2\right\}. \tag{34}$$

The final task in the problem is to summarise the posterior by calculating point and interval estimates, which we will describe next.

### 3.3.1 Laplace approximation to the posterior

One of the most commonly used approximations to the posterior is the Laplace approximation, in which the posterior density is approximated with a Gaussian distribution (MacKay, 2003). The mean of the distribution is set to the *maximum a posteriori* (MAP) estimate, and the covariance is calculated based on the curvature of the posterior around the MAP estimate, i.e., $\pi(\tilde{\mathbf{N}}|\mathbf{y}) \approx \mathcal{N}(\tilde{\mathbf{N}}_{\mathrm{MAP}}, \tilde{\Gamma}_{\mathrm{post}})$. The MAP estimate is defined as the point where the posterior density has a maximum:

$$\tilde{\mathbf{N}}_{\mathrm{MAP}} = \underset{\tilde{\mathbf{N}} \in \mathcal{D}}{\arg\max}\, \pi(\tilde{\mathbf{N}}|\mathbf{y}), \tag{35}$$

where $\mathcal{D} \subset \mathbb{R}^{n_B}$ is a space of possible values for $\tilde{\mathbf{N}}$. In this paper, we use the Gauss-Newton algorithm equipped with line search to compute the MAP estimate. The posterior covariance is given by

$$\tilde{\Gamma}_{\mathrm{post}} = \left(J(\tilde{\mathbf{N}}_{\mathrm{MAP}})^T \tilde{\Gamma}_e^{-1} J(\tilde{\mathbf{N}}_{\mathrm{MAP}}) + \tilde{\Gamma}_{\mathrm{pr}}^{-1}\right)^{-1}, \tag{36}$$

where $J(\tilde{\mathbf{N}}_{\mathrm{MAP}})$ denotes the Jacobian matrix of $h(\tilde{\mathbf{N}})$ with respect to $\tilde{\mathbf{N}}$, evaluated at the MAP estimate.

The Laplace approximation to the posterior covariance can be used to compute parameter uncertainty estimates such as approximate credible intervals (CI), which in the Bayesian framework can be directly interpreted as statements on the probabilities of the parameter values. This means that the true parameter value is contained within the $p$-% CI with $p$-% probability. The CI does not have a unique definition, but here for the Laplace approximation we define a 95 % credible interval as $\tilde{\mathbf{N}}_{\mathrm{MAP}} \pm 1.96\tilde{\sigma}_{\mathrm{post}}$, where $\tilde{\sigma}_{\mathrm{post}}^2(i) = \tilde{\Gamma}_{\mathrm{post}}(i,i)$, i.e. the values on the diagonal of the posterior covariance matrix. Estimates for $\mathbf{N}$ can be calculated by transforming the log-parametrized PSD values to the absolute scale as $10^{\tilde{\mathbf{N}}_{\mathrm{MAP}} \pm 1.96\tilde{\sigma}_{\mathrm{post}}}$. Note that due to the parametrization the distribution of $\mathbf{N}$ is not Gaussian but log-normal.





### 3.3.2 Full characterization of the posterior

In many cases, the MAP estimate and Laplace approximation are sufficient to summarise the posterior and quantify uncertainty
in the parameters. However, to properly assess the validity of this approximation the posterior needs to be fully characterized.
In practice, this is done by sampling methods such as Markov chain Monte Carlo (MCMC), which generate correlated samples
$\{\tilde{\mathbf{N}}_i, i = 0, 1, \ldots\}$ that are distributed according to the posterior probability density. The basic algorithm to construct a MCMC
sampler is the Metropolis algorithm (Metropolis et al., 1953), where a proposal $\tilde{\mathbf{N}}'$ for a new sample is generated based on
the current sample $\tilde{\mathbf{N}}_i$, and the proposal is accepted with a probability of the posterior ratio, $\min(1, \pi(\tilde{\mathbf{N}}'|\mathbf{y})/\pi(\tilde{\mathbf{N}}_i|\mathbf{y}))$. If the
285 proposal is accepted, we set $\tilde{\mathbf{N}}_{i+1} = \tilde{\mathbf{N}}'$ and otherwise $\tilde{\mathbf{N}}_{i+1} = \tilde{\mathbf{N}}_i$. The algorithm has to be run for long enough so that the
chain is converged, i.e. will have properly sampled the posterior and sufficiently accurate estimates can be computed. How long
is long enough depends on many factors, such as the required accuracy, efficiency of the sampler, and shape of the posterior.
Assessing convergence reliably can be difficult, but is a requirement before any conclusions from the samples can be drawn.

Once a set of samples representative of the posterior has been obtained, estimates of the conditional mean (CM) and credible
intervals can be computed. The CM estimate is the expected value of the posterior, i.e. its center of mass, and is computed as

$$\tilde{\mathbf{N}}_{\mathrm{CM}} = \int_{\mathcal{D}} \tilde{\mathbf{N}} \, \pi(\tilde{\mathbf{N}}|\mathbf{y}) \, \mathrm{d}\tilde{\mathbf{N}} \approx \frac{1}{n_s} \sum_{i=1}^{n_s} \tilde{\mathbf{N}}_i, \tag{37}$$

where $n_s$ is the number of samples. In the case of MCMC we define the $p$-% credible interval as the interval between the two
ends, or *tails*, of the marginal posterior distribution, where both tails contain $(100 - p)/2$ % of the marginal posterior samples.
The true shape of the marginal posteriors can also be visualized by histograms of the samples.

Drawing lots of samples to approximate the posterior is naturally slower than computing the Laplace approximation, and the
feasibility of applying MCMC on a given problem depends highly on the efficiency of the chosen sampler. In this work, we
use the Metropolis-adjusted Langevin algorithm (MALA) (Roberts and Tweedie, 1996) with adaptive proposals (Haario et al.,
2001). Compared to a regular random walk Metropolis algorithm, MALA increases sampling efficiency by using the gradient
of the posterior to guide the proposals towards areas of higher posterior probability. For more information on MCMC methods,
see for example Brooks et al. (2011).

### 3.4 Uncertainties in the forward model

In addition to the unknown of primary interest, $\tilde{\mathbf{N}}$, the forward model includes auxiliary (sometimes also called nuisance)
parameters, ones that we are not primarily interested in but nevertheless affect the model output. These include parameters
such as exhaust flow velocity, fractal dimension, and the Hamaker constant. Above, the inference of the PSD was written
assuming that these parameters are known exactly. This, of course, is seldom the case in reality and the assumption may lead
to erroneous results and unrealistically narrow uncertainty estimates. To quantify the effect of uncertainty in the auxiliary
parameters on the posterior probability of $\tilde{\mathbf{N}}$, we can use marginalization, where the auxiliary parameters are modelled as
additional unknowns and then integrated out.



Let us denote the auxiliary unknowns by $\nu \in \mathbb{R}^{n_\nu}$, and define a prior $\pi(\nu)$ that is independent of $\pi(\tilde{\mathbf{N}})$. Then, the augmented form of the posterior reads

$$\pi(\tilde{\mathbf{N}}, \nu | \mathbf{y}) \propto \pi(\mathbf{y} | \tilde{\mathbf{N}}, \nu) \pi(\tilde{\mathbf{N}}) \pi(\nu). \tag{38}$$

By carrying out the following integral:

$$\pi_\nu(\tilde{\mathbf{N}} | \mathbf{y}) = \int\limits_{\mathbb{R}^{n_\nu}} \pi(\tilde{\mathbf{N}}, \nu | \mathbf{y}) d\nu, \tag{39}$$

the effect of $\nu$ is integrated out and we are left with a marginal posterior probability $\pi_\nu(\tilde{\mathbf{N}} | \mathbf{y})$. We use the subscript $\nu$ to denote the variable(s) over which the posterior is marginalized, to avoid confusion with Eq. (34) which implicitly considers $\nu$ known. In practice, the integration is done by carrying out MCMC sampling for both $\tilde{\mathbf{N}}$ and $\nu$ simultaneously. Once a representative set of samples has been collected, the marginalization (39) is trivial.

## 4    Case studies

Let us now use the described inversion approach to analyze measurements of particle emissions from fuel-operated auxiliary heaters (AH) used in vehicle preheating or providing additional heat (Oikarinen et al., 2022). In this kind of measurement, often i) the PN concentration is high, and ii) the sample needs to be transported some distance from the end of the exhaust to the measurement device; two factors that increase the likelihood of coagulation and wall losses distorting the PSD, and question the validity of the measurements unless these effects are tested and corrected for. We will compute both the Laplace approximation and a full characterization of the posterior with MCMC to compare these approaches. Two cases will be studied, one based on synthetic data where the true initial PSDs are known, and one considering real measurements.

### 4.1    Modelling the measurement setup

The PSD measurements in Oikarinen et al. (2022) were carried out using the Engine Exhaust Particle Sizer™ (EEPS) model 3090, manufactured by TSI Inc. The EEPS is a spectrometer that estimates PSD based on measurements of electrical mobility. A simplified schematic of the measurement setup in Oikarinen et al. (2022) is shown in Fig. 3, where exhaust gas from the AH is led to the EEPS through a sampling line with a length of 3.2 m and an inner diameter of 5 mm. Flow velocity through the line was 3.5 m/s, leading to a residence time of approximately 0.9 seconds. The sampling line was heated to 250 °C, and just before the EEPS the sample was diluted at a ratio that varied between 40:1 and 100:1 to allow for measurements of PN concentrations higher than the upper limit of the instrument.

The operating principle of the EEPS is, in short, the following (TSI., 2015): A flow of exhaust gas is directed at the inlet, where particles in the gas are positively charged. The charged particles are then directed in a laminar flow through a cylindrical tube that has a positive high-voltage electrode column in the center and a stack of 22 circular electrometers on the outer wall. The electric field between the center electrode and electrometers deflects the particles outwards so that particles with high electrical mobility hit the electrometers near the inlet and particles with low electrical mobility hit the electrometers further





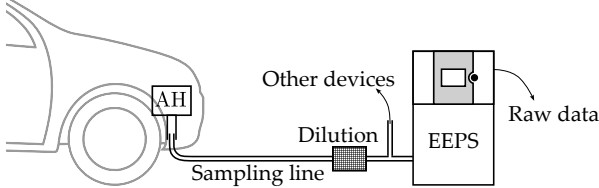

**Figure 3.** A schematic of the measurement setup.

along the stack. The striking particles generate currents that are proportional to the particle concentration. Measurements of
these currents with their location along the stack can be inverted to give an estimate of the PN concentration. The operating
size range of the EEPS is between 5.6 nm and 560 nm, and the whole size range is measured simultaneously, every one second
in the present case.

An observation model (cf. equation (32)) for the EEPS in terms of the raw electrometer currents can be written as (Wang
et al., 2016a):

$$\mathbf{y} = \mathbf{H}\mathbf{f} + \mathbf{e}, \tag{40}$$

where $\mathbf{y} \in \mathbb{R}^{22}$ are the measured currents, $\mathbf{f} \in \mathbb{R}^{17}$ denotes a representation of the size distribution used by the EEPS, and
$\mathbf{e} \in \mathbb{R}^{22}$ is measurement noise. The size distribution $\mathbf{f}$ is mapped to currents with an instrument matrix[2] $\mathbf{H} \in \mathbb{R}^{22 \times 17}$, which
can be exported from the EEPS[3]. Here we use the SOOT instrument matrix (Wang et al., 2016b), which TSI has developed
to give more accurate estimates with soot particles than the instrument's default instrument matrix. To connect the aerosol bin
model (3) to the observation model above, we map the size bin representation to $\mathbf{f}$ with an interpolation matrix $\mathbf{B} \in \mathbb{R}^{17 \times n_B}$:

$$\mathbf{f} = \mathbf{B}\hat{\mathbf{N}}, \tag{41}$$

where $\hat{\mathbf{N}}$ is a size bin representation of the PSD of the sample when it enters the EEPS, after it has been altered by the sampling
line. Finally, let $\Xi$ represent the coagulation and wall diffusion model so that $\hat{\mathbf{N}} = \Xi(\mathbf{N})$, and let $\phi$ denote a transformation
from the log-space to the absolute scale $\mathbf{N} = \phi(\tilde{\mathbf{N}})$. The full observation model is then $\mathbf{y} = h(\tilde{\mathbf{N}}) + \mathbf{e}$, where the forward model
is

$$h(\tilde{\mathbf{N}}) := \mathbf{H}\mathbf{B}\,\Xi(\phi(\tilde{\mathbf{N}})). \tag{42}$$

To specify the covariance of the measurement noise, $\Gamma_e$, we assume that noise in each electrometer is independent of the
other electrometers, so that $\Gamma_e$ is diagonal and consists of the noise variances $\sigma_{e,i}^2$ of each electrometer $i$. These variances can
generally be estimated from a calibration measurement with a minimal number of particles using a filtered sample. However,
an analysis of the measurement data shows in this case that the noise seems to be approximately proportional to the measured

---

[2]Also known as an *inversion matrix* in the EEPS manual.

[3]If the raw current data and/or the instrument matrix were *not* available, inversion could also be done based on the EEPS estimate of the PSD. However,
specifying the distribution of the uncertainty, or noise, in the data would not be as straightforward as with the raw data.



**Table 1.** True values for the auxiliary parameters that were used to simulate synthetic data, and the range over which they are allowed to vary when marginalizing.

| Parameter | True value | Prior range |
| --- | --- | --- |
| Temperature $T$ | 250 °C | 230 – 270 °C |
| Fractal dimension $D_f$ | 1.7 | 1.5 – 2.2 |
| Primary particle diameter $d_1$ | 27 nm | 20 – 34 nm |
| Hamaker constant $A$ | 2.0 ($10^{-19}$ J) | 1.0 – 5.0 ($10^{-19}$ J) |
| Flow velocity $v_f$ | 3.5 m/s | 3.2 – 3.8 m/s |

current. Specifically, during short enough time windows, where the PSD is expected to stay constant and thus variation in the measured signal can be attributed to noise, standard deviation of the measurements is roughly 5 - 15 % of the amplitude of the measurement. An exception to this is when the current drops below the electrometer noise floor. We therefore model the noise variance as

$$\sigma_{e,i}^2 = \max\left(\sigma_{\text{floor},i}^2, (\epsilon y_i)^2\right), \quad i = 1, \ldots, 22, \tag{43}$$

where $\sigma_{\text{floor},i}^2$ and $y_i$ denote the noise floor and measured current of electrometer $i$, respectively, and $\epsilon$ denotes the noise level. Note that in the derivation of the likelihood we assumed that the noise and the unknowns are independent, but here some dependency is introduced between them. In practice, we thus use the measured values of $y_i$ to assess the variance in Eq. (43) instead of reformulating the likelihood model to accommodate for a non-additive noise model that would result from the dependency between $N$ and noise. The effect of this approximation, however, is minor if the noise level is not very high.

### 4.2 Inversions with simulated data

We created simulated data with the above model (Eq. (42)), specifying the initial PSD and auxiliary parameters, and adding noise with level $\epsilon = 0.10$. The initial number concentration was chosen to be high enough to clearly show coagulation effects, but still realistic to be comparable to the number concentrations seen in the real measurements shown later. To avoid an inverse crime (Kaipio and Somersalo, 2006), the data were created using a higher number of size bins than what was used in the inversion. We used 30 and 16 bins per decade for the data generating and inversion models, respectively. The dilution ratio was set to 100:1, and the auxiliary parameters were set to the values given in Table 1. The expected value of the prior was set to $\tilde{\mathbf{N}}_* = \log(10^3/w_{\text{bin}})$, where $w_{\text{bin}}$ is the logarithmic bin width, and correlation length and variance were set to $\tilde{l} = 12/16$ and $\tilde{a} = 4$, respectively.

#### 4.2.1 Known auxiliary parameters

First, we carried out inversion with the same auxiliary parameters that were used to create the data. In this case, the only source of modeling error with respect to the accurate model used for generating the synthetic data is that caused by the coarser





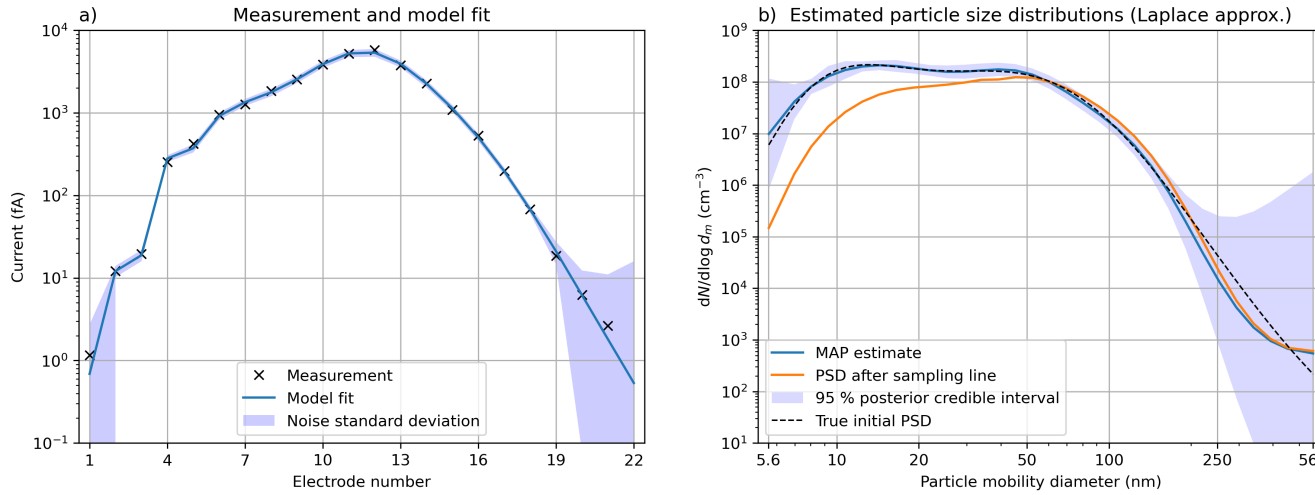

**Figure 4.** Synthetic data. a) Simulated noisy data, model fit corresponding to the MAP estimate, and standard deviation of the measurement noise. b) MAP estimate and 95 % posterior credible interval of the initial PSD, the MAP estimate propagated through the sampling line, and the true initial PSD.

discretization of the particle size. The specified initial PSD, and the corresponding generated noisy data points are shown in Fig. 4.

The Gauss-Newton algorithm to compute the Laplace approximation for the initial PSD converged in 9 iterations and took 0.4 seconds. All calculations in this paper were done on a laptop with an Intel(R) Core(TM) i7-12700H processor, using only a single CPU core. The Laplace approximation for the initial PSD and the PSD after the sampling line are also shown in Fig. 4. The PSD after the sampling line is calculated from the MAP estimate with the coagulation and wall diffusion model, and depicts what would be estimated based on the EEPS data, if the effects of the PSD evolution within the sampling line were

neglected. A comparison between the estimated PSDs before and after the sampling line reveals a clear coagulation effect; concentration of the smallest particles has reduced by up to two orders of magnitude, and these particles show up as slightly higher number concentration of larger particles. The initial PSD can be said to be estimated well because the true initial PSD is found within the 95 % CIs over the whole size range. The relative error between the MAP estimate and the true initial PSD is 6.2 %.

To compute the true posterior, we ran the MCMC sampler for 200,000 iterations which took 20 minutes. The first 25 % of the samples were removed as burn-in, and the remaining 150,000 samples had an average integrated autocorrelation time of 20.2 samples which means there were over 7,400 independent posterior samples in the chain. Figure 5 a)-c) shows the first variable of the MCMC chain (i.e., the smallest size bin) as an example, and a visual inspection of this and the rest of the chains showed no sign of convergence issues. In the remainder of this paper, it is assumed that a similar amount of sampling and a

similar convergence analysis is done to validate each presented MCMC result. Figure 5 d) shows the CM estimate and 95 % CIs. Again, the initial PSD is estimated well and is found within the CIs. The relative error between the CM estimate and the





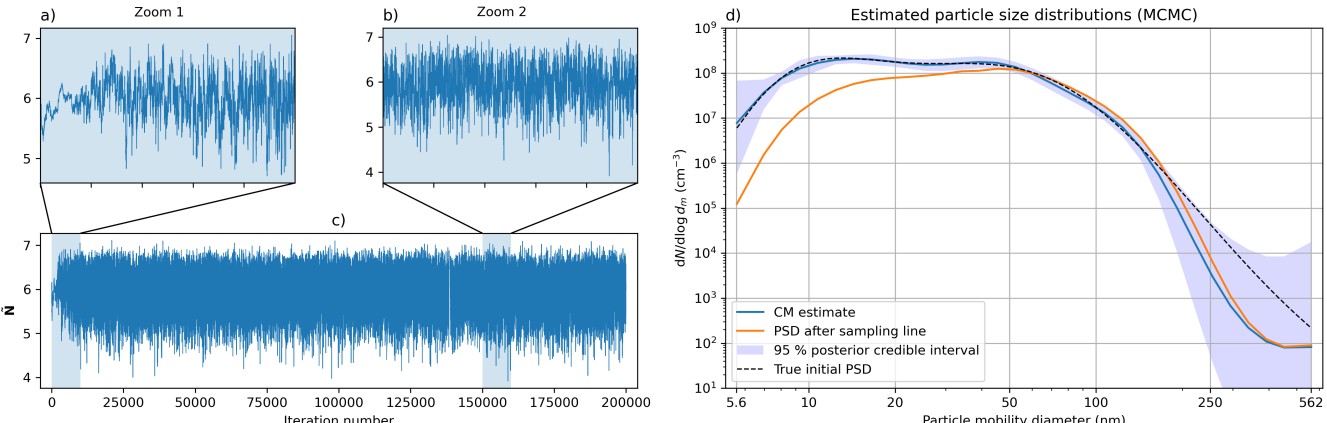

**Figure 5.** Synthetic data. The full MCMC chain, shown in c), of the $\log$ number concentration of the first size bin, and two zoomed in sections showing a) the beginning of the chain which is still in burn-in, and b) part of the chain when it has reached the steady state. d) CM estimate and posterior credible intervals calculated with MCMC using the true auxiliary parameters.

true initial PSD is 7.1 %. The main differences compared to the Laplace approximation are that for particles larger than 200 nm, the CM estimate gives lower concentrations than the MAP estimate, and that the CIs obtained by sampling are narrower than those of the Laplace approximation.

### 4.2.2 Unknown auxiliary parameters

As mentioned in Sect. 3.4, it not always realistic to assume the auxiliary parameters are known accurately, and one approach to take their uncertainty into account is to model the auxiliary parameters as additional unknowns. An example of what can happen when just one parameter, here the fractal dimension, is incorrectly specified and its uncertainty not modelled, is shown in Fig. 6. Here the data were generated with $D_f = 1.7$, but in the inversion we used $D_f = 2.1$ which is well within the possible values for fractal dimension of soot. Figure 6 a) shows, that although not a dramatic change from the results in Fig. 5, the initial PSD is underestimated and the 95 % credible intervals do not always contain the truth, especially around 10 nm. In addition, the relative error between the CM estimate and the truth has increased to 18.8 %.

Let us now model $D_f$ as an unknown and marginalize it. The resulting posterior is shown in Fig. 6 b). Compared to the model where $D_f$ was assumed to be accurate, the CIs of especially particles under 50 nm (which seem to be the most affected by coagulation) are considerably wider, and now do include the true initial PSD. The CM estimate is also closer to the truth, with the relative error being 10.1 %.

Finally, let us marginalize all other auxiliary parameters as well, one by one and also all together. A complete list of the auxiliary parameters is given in Table 1, which also lists the ranges over which the integration is carried out. The fixed auxiliary parameters are held at their true values, except for $D_f$, which is set to 2.1. Figure 7 a) shows the full posterior in the case where all auxiliary parameters were integrated out at the same time. As is expected, the CIs in this case are wider than when only



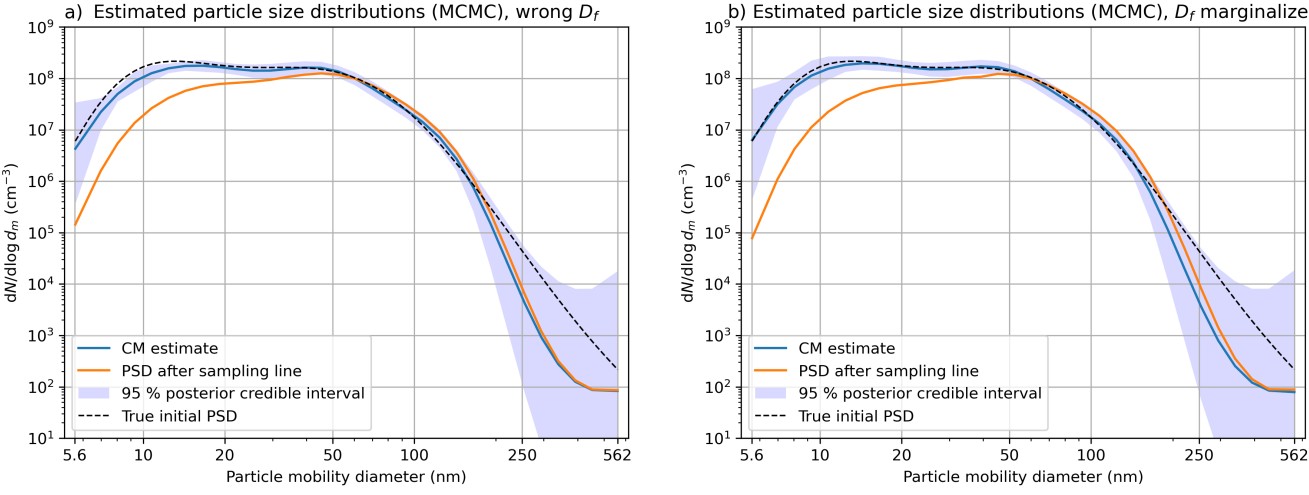

**Figure 6.** Synthetic data. a) Posterior distribution with fractal dimension set to an incorrect value. b) Posterior distribution with fractal dimension marginalized.

the fractal dimension has been marginalized. The relative error between the CM estimate and the truth is 6.8 %, which is even lower than when using the correct auxiliary parameters (7.1 %). However, this is likely just a coincidence related to, e.g., the chosen parameters ranges for marginalization, and not something to be expected in general.

To visualize the effects of uncertainties in different auxiliary parameters, let us focus on the posterior of a single size bin, i.e. a *marginal* posterior, which can be plotted as a histogram. Figure 7 b) shows these marginal posteriors at the 6th size bin (corresponding to $d_m = 12.40$ nm), of five cases, where either the flow velocity, Hamaker constant, fractal dimension, all, or none of the auxiliary parameters have been marginalized. Note that the particle number in Fig. 7 b) is shown on the $x$-axis, and the $y$-axis now denotes the posterior probability density. Marginalization of temperature or primary particle diameter is not shown because their effect on the posterior was minimal. Over the considered integration ranges, the effects of fractal dimension and the Hamaker constant were larger than that of the flow velocity, but in each case the marginalization increased the width of the CIs. Marginalizing all auxiliary parameters at the same time had naturally the largest effect.

### 4.3 Fuel-operated auxiliary heater measurement

The measurement campaign by Oikarinen et al. (2022) was carried out in February 2021 in Finnish winter conditions (-19 to -7 °C) over several days and with multiple vehicles. For the scope of this paper, we chose a measurement of one of the vehicles, a gasoline-powered 2019 Volkswagen Golf, to analyze as an example. The vehicle had an OEM installed AH manufactured by Webasto Ltd. Emissions of the AH were measured over a half hour period, which included a cold start and shutdown of the AH. The EEPS was set to measure at one second intervals, leading to a total of around 1900 measurements. A more detailed description of the setup and analysis of the EEPS-inverted data is found in the original paper.





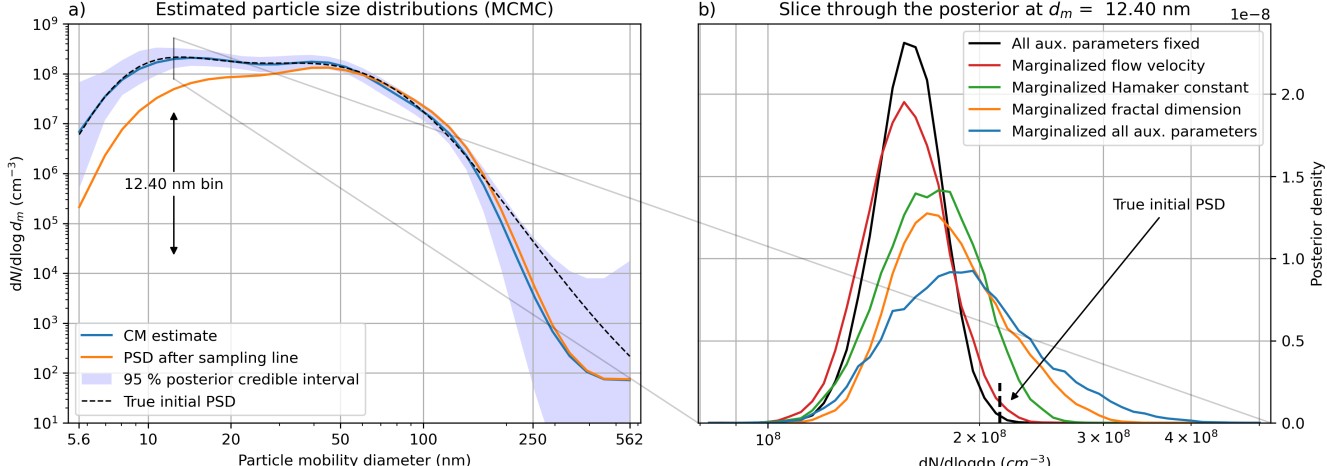

**Figure 7.** Synthetic data. a) Posterior with all auxiliary model parameters marginalised. b) Marginal posteriors of size bin number 5 (12.40 nm) with different auxiliary parameters marginalized.

As mentioned earlier, computing the Laplace approximation is much faster than doing MCMC sampling and was therefore
used to invert the dataset. MCMC was carried out at a few selected measurement points, discussed later. Discretization of the inversion model was set to 16 bins per decade, so that the output bins were the same as those of the EEPS. The prior parameters were also kept the same as in the synthetic data inversion. However, because the choice of prior is subjective to some degree, in Sect. 5 we will briefly discuss different choices for the prior parameters and their influence on the PSD estimates. Further, because the data exhibit temporal smoothness, we used the previous MAP estimate as an initial guess for computing the next
one, which reduced the number of Gauss-Newton iterations required to minimize the cost function. Inverting the whole dataset took 2 minutes and 20 seconds.

The MAP estimates of the initial PSD for the duration of the whole measurement are shown in Fig. 8. The results are qualitatively similar to the ones obtained directly from the EEPS, discussed in Oikarinen et al. (2022), where after startup, the AH emissions are relatively stable, followed by a burst of sub-20 nm particles during shutdown. However, there is a large
difference in the total PN. Compared to the (diffusion-corrected) EEPS estimates, the total PN is on average 50 – 100 % higher in the MAP estimates, varying over the measurement period with a peak of 230 % higher during the AH shutdown. Because coagulation affects the particle population differently depending on the particle size, also the ratio of PN between the MAP and EEPS estimates depends on the particle size. This is visualized in Fig. 9, which shows the ratio of PNs in the MAP and EEPS estimates for a few ranges of particles sizes. The number of particles under 20 nm may be over 3 times higher with the
sampling line modelled, whereas the number of particles over 100 nm is typically reduced by up to 50 %.

To compare the MAP estimates and EEPS output in more detail, let us analyze the two time instants shown as red dashed lines in Fig. 8. The time instant at Line 1 in the middle of the measurement corresponds to the steady state and the time instant at Line 2 at the end is the time when the burst of sub-20 nm particles was at its peak. The Laplace approximations




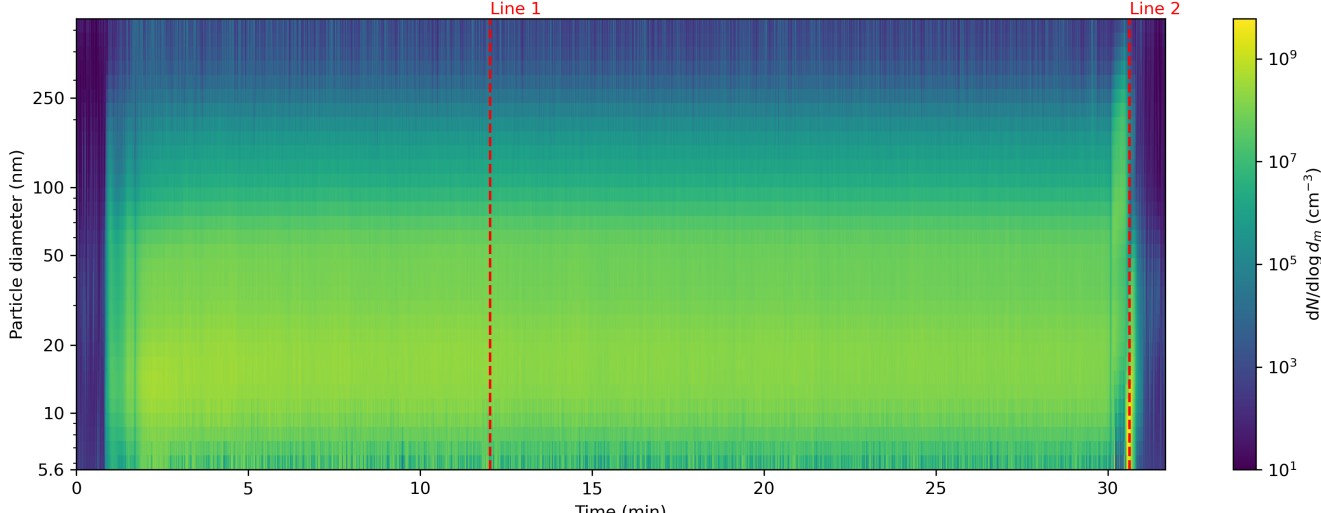

**Figure 8.** The MAP estimates for the initial PSD for a measurement of AH exhaust emissions of a 2019 Volkswagen Golf. Lines 1 and 2 denote measurement times analyzed in more detail.

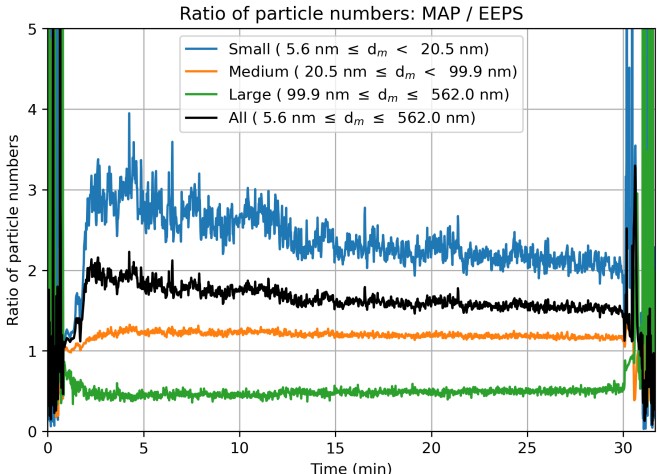

**Figure 9.** Ratio of PNs between the MAP and diffusion-corrected EEPS estimates during the example measurement. The MAP estimate of the number of small particles is three times that of the EEPS estimate in the in the beginning, whereas the number of particles over 100 nm is only around half of that of the EEPS estimate.

and the corresponding data fits at these points are shown in Figs. 10 and 11. First, in both examples, modelling the sampling line increases the number of the smallest particles in the estimates significantly while reducing the number of larger particles. Second, the PSDs after the sampling line are close to the PSD estimates from the EEPS, as they should be if our inversion






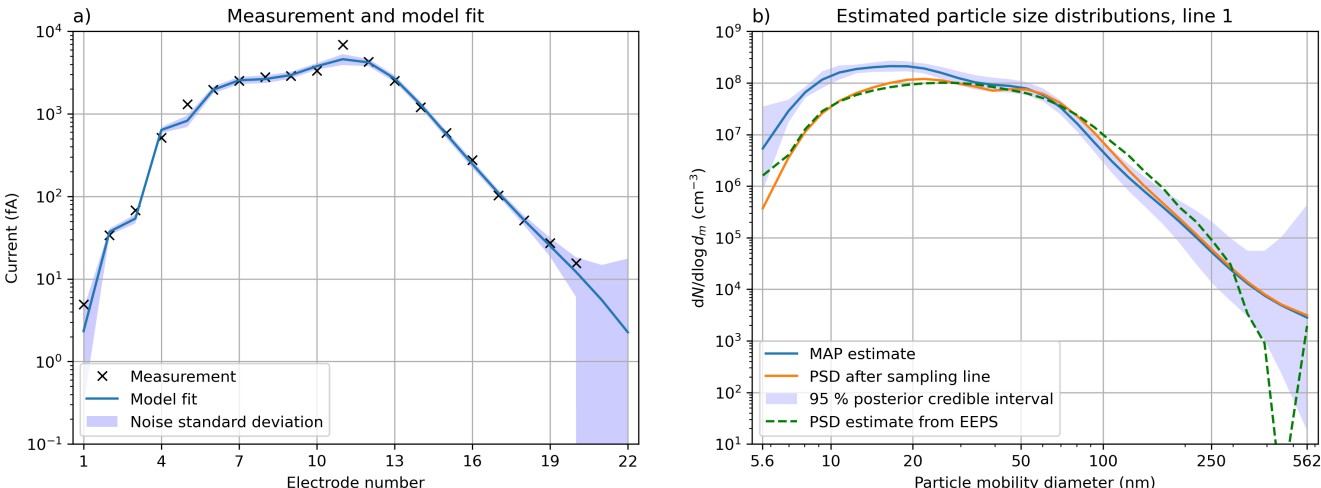

**Figure 10.** AH measurement at time instant 1 (Line 1 in Fig. 8). a) Measured data, model fit corresponding to the MAP estimate, and standard deviation of the measurement noise. b) MAP estimate and 95 % posterior credible interval of the initial PSD, the MAP estimate propagated through the sampling line, and the PSD estimate given by the EEPS.

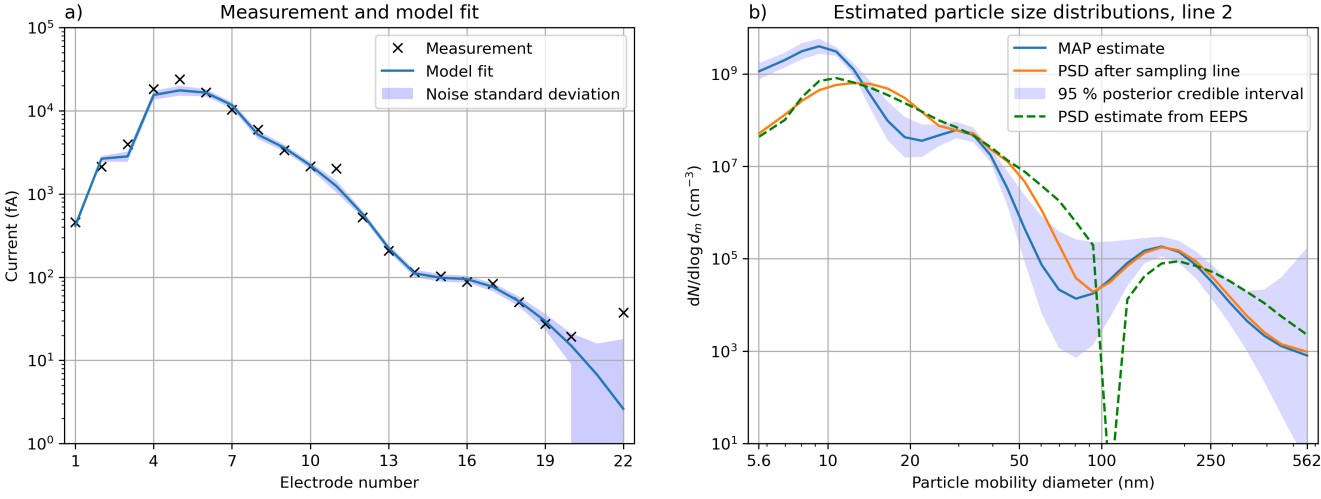

**Figure 11.** AH measurement at time instant 2. Refer to the caption of Fig. 10.

algorithm was working correctly. The differences between them can be mostly attributed to the different inversion algorithms, choices of prior/regularization parameters and handling of measurement noise. Inversion in the EEPS is based on a Tikhonov-regularized Gauss-Markov least squares solution (Mirme et al., 2007; Mirme and Mirme, 2013; Wang et al., 2016a), but its

details are not public. Moreover, the EEPS sometimes rounds the PSD estimates of some size bins down to zero, which can be seen in both Figs. 10 and 11 (note that we have replaced the zero-values by ones for a better plot on a log-scale).



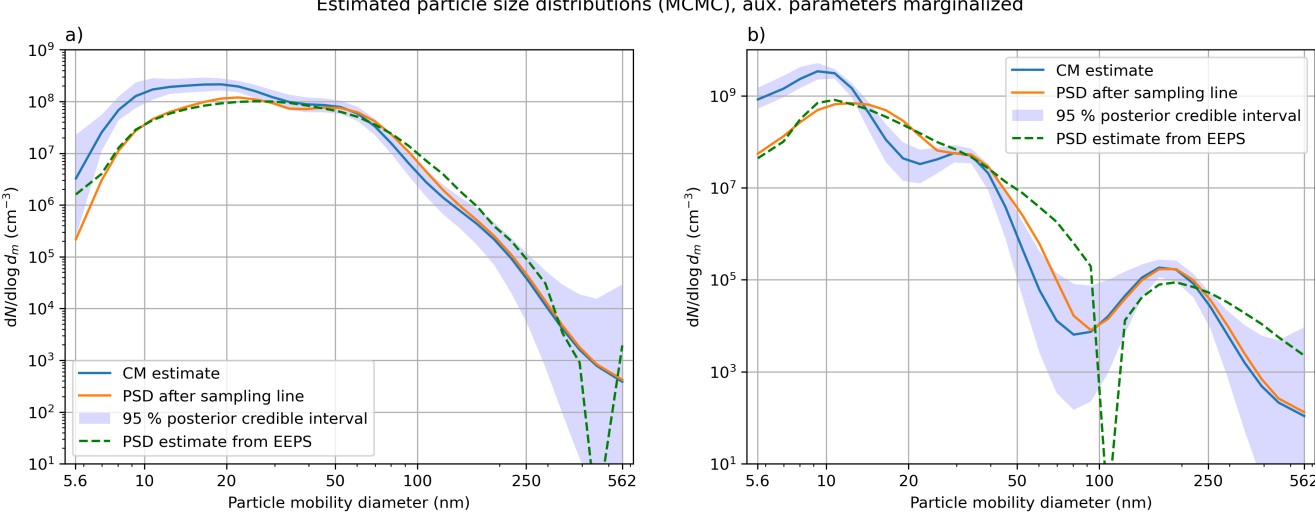

**Figure 12.** AH inversion results with all auxiliary parameters marginalized using MCMC for a) Line 1 and b) Line 2.

Finally, we used MCMC to marginalize the auxiliary model parameters for the measurements at Line 1 and 2. The marginalization was carried out simultaneously for all five parameters over the ranges listed in Table 1. The results, shown in Fig. 12, resemble those of the synthetic tests in that the CM and MAP estimates are still close, but the 95 % CIs in the marginalized case are wider especially for the smallest particles.

## 5 Discussion

The analyzed synthetic and real measurements show that sampling lines can have a major effect on the PSD when measuring the emissions of fuel-operated auxiliary heaters. The same observation obviously applies to any measurement where the sampling line is long and/or the PN concentration is high. When should one then be concerned about their measurements being distorted, and consider applying, for example, the methods in this paper? The answer depends on not just the sampling line and PN concentration, but also on factors such as the particle shape, density, fluid they are suspended in etc., which affect the coagulation rate and particle diffusion. In the context of the specific numerical example studied in *this* paper (see Fig. 4), our model shows that if the length of the sampling line was reduced from 3.2 meters to, say, one meter, the true initial total PN between 5.6 nm and 562 nm would still be about 33 % higher than what would be measured after the sampling line. It could be possible to define an indicator for a risk of distorted measurements based on the residence time and measured PN, but this is out of the scope of the present study.

A question regarding the inversion is if there is enough benefit to sampling the true posterior with MCMC versus just computing the Laplace approximation. Based on the results in Sect. 4, the answer is in most cases no; the MAP and CM estimates are similar, and the Gaussian density seems to approximate the posterior well. And most importantly, computing



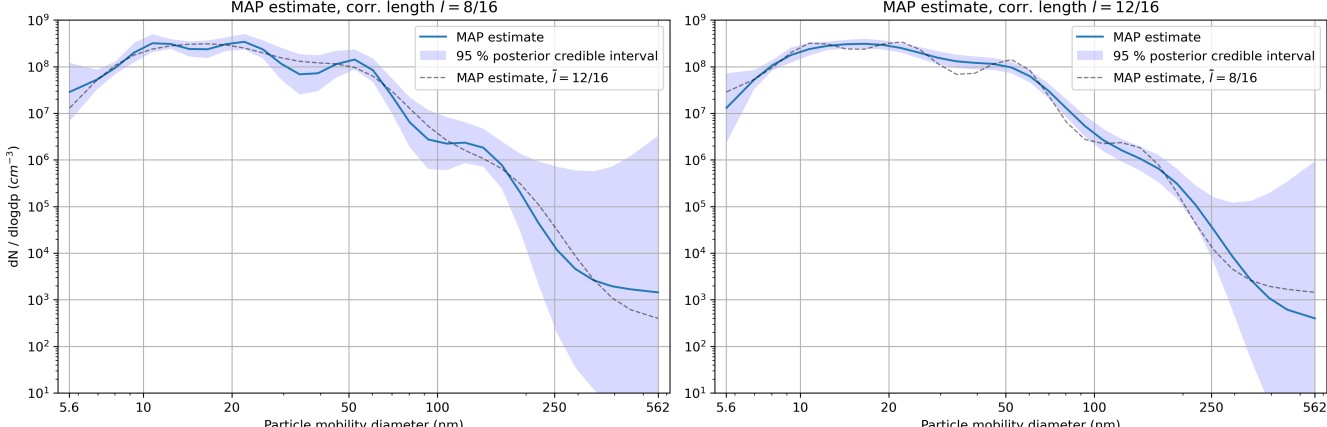

**Figure 13.** The influence of prior correlation length shown as MAP estimates and 95 % CIs with a) $\tilde{l} = 8/16$, and b) $\tilde{l} = 12/16$. The overlaid dotted lines are the MAP estimates from the neighboring figure for an easier comparison.

the Laplace approximation is fast enough that it could even be combined with the inversion algorithm in the EEPS and still produce the estimates in real time, only now taking into account the effect of the sampling line and providing uncertainty estimates as well. A major benefit of running MCMC is that parameters in the forward model that we are uncertain of can be integrated out, and their uncertainty propagated to the PSD estimates. The trade-off is an increased computation time. However, it could be possible to account for this uncertainty in an approximative manner while still retaining the speed of the Laplace

approximation, by using the Bayesian approximation error method (Kaipio and Somersalo, 2007; Kaipio and Kolehmainen, 2013), but this hasn't been pursued in this work.

As mentioned earlier, the inverse problem of estimating the PSD from measurements of currents is highly ill-posed even without considering the sampling line, and hence the uncertainty of the posterior depends to a large extent on the prior assumptions. For example, as the correlation length $\tilde{l}$ (see Sect. 3.1) is increased, the posterior will be constrained to more and more

smooth solutions, which also constrains the posterior uncertainty. Let us demonstrate this in Fig. 13 by plotting the Laplace approximations of two posteriors which differ only in the prior correlation length. Note how in addition to smoothing out the MAP estimate, increasing the correlation length also decreases the width of the credible intervals, i.e. decreases the posterior uncertainty.

A source of uncertainty related to coagulation that hasn't been modelled in this work is the influence of particles larger

and smaller than what we can measure, $> 562$ nm and $< 5.6$ nm, respectively, in the case of the EEPS. In the measurable range, coagulation with these larger particles results in higher than modelled particle losses, whereas coagulation with the smaller particles both introduces new particles at the lower measurable limit as well as causes particles in the measurable range to grow faster than modelled. Hence, the presence of particles larger than we can measure will lead to underestimating the initial PSD and the presence of particles smaller than we can measure will have the opposite effect. According to a numerical



simulation over a wide size range, however, if we assume the PSD outside the measurable range to continue the downward trend that is seen, for example, in Fig. 12 a), its effect on the measured PSD is negligible.

Finally, there are a few loss mechanisms, including sampling losses (extraction of the aerosol at the sampling location), gravitational losses, inertial impaction, and thermophoresis, that may modify PN concentrations in a sampling line (Giechaskiel et al., 2012), but that are likely negligible in the case studied here and therefore not modelled. Sampling losses, gravitational

losses, and inertial impaction mainly affect particles $> 1 \ \mu$m, which is larger than what the EEPS can measure. On the other hand, inertial impaction can also cause losses of nanometer-size particles, but these are at or below the measured size range. Thermophoresis, the motion of particles in a temperature gradient, is potentially a major loss mechanism in sampling lines if there is a large temperature difference between the exhaust gas and the sampling line walls. In the measurement by Oikarinen et al. (2022), the sampling line was heated to 250 °C which is very close to the temperature of the exhaust gas exiting the AH

exhaust pipe, and hence the thermophoretic losses are expected to be minimal.

## 6  Conclusions

In this article, we investigated how coagulation and wall diffusion in a sampling line modify a sample's particle size distribution and hence bias its measurement. Coagulation was found to become a significant factor especially at high PN concentrations, because while the rate of wall diffusion is independent of the number concentration the rate of coagulation is approximately

proportional to its square. As an example of a high PN case we studied the exhaust emissions of a fuel-burning road vehicle auxiliary heater. Through simulations and examining real data we found that, in a typical measurement setup, coagulation in the sampling line can reduce the total PN by more than 50 %, and the number of small particles ($< 20$ nm) even by an order of magnitude. The initial particle size distribution, not yet biased by the sampling line, can however be estimated from measurements that were done after the sampling line, given that we have a model for the processes in the sampling line.

This is an ill-posed inverse problem which we solved using methods in the Bayesian framework for inverse problems. In this framework, we can incorporate prior information about the size distribution and carry out a systematic calculation of the uncertainty related to the estimates. Two approaches to exploring the posterior probability density were tested: one where the posterior is fully characterized by MCMC sampling, and another where the posterior is approximated as a Gaussian distribution. These resulted in similar estimates for the initial PSD, but both approaches also had distinct advantages. With the Gaussian

approximation the inverse problem could be solved in a computationally efficient manner, so that computing the estimates of the initial size distribution and its uncertainty could be done in real time even with measurement devices that have fast response times ($1 - 10$ Hz, for example). Using MCMC, on the other hand, we were able to take into account the (possible) uncertainty in the parameters of the coagulation and diffusion models, such as particle shape, by using marginalization. This results in uncertainty estimates that are more consistent with what we assume to know about the problem, but are computationally more

demanding to calculate.

In conclusion, when the number concentration is high and the particles are small, the effect of coagulation in sampling lines should be taken into account when carrying out PSD or PN measurements. Otherwise, the particle numbers may be severely



underestimated. Further, estimates of the PSDs have major uncertainties associated with them that traditional, deterministic, inversion methods do not convey. Bayesian methods for uncertainty quantification could therefore prove useful in assessing the

health and environmental impacts of fine particles in the future.

*Code and data availability.*   The current version of SLIC is available from the project website: https://github.com/mniskanen/sampling-line-inversion under the MIT licence. The exact version of the model as well as input data and scripts to run the model and to produce the results and plots for all the simulations presented in this paper are archived on Zenodo: https://zenodo.org/doi/10.5281/zenodo.12188947.

*Author contributions.*   MN: formal analysis, investigation, methodology, software, validation, visualization, writing - original draft prepara-
tion, AS: methodology, writing - review & editing, HO: conceptualization, data curation, writing - review & editing, MO: conceptualization, data curation, investigation, writing - review & editing, PK: conceptualization, funding acquisition, project administration, supervision, writing - review & editing, SM: conceptualization, funding acquisition, project administration, supervision, writing - review & editing, KL: conceptualization, funding acquisition, methodology, project administration, writing - review & editing

*Competing interests.*   The authors declare that they have no conflict of interest.

*Acknowledgements.*   This research has been supported by the Research Council of Finland Center of Excellence project VILMA (grant no. 346375), the Jane ja Aatos Erkon Säätiö (project AHMA), the Tampere Institute for Advanced Study (Tampere IAS), the Academy of Finland competitive funding to strengthen university research profiles (PROFI) for the University of Eastern Finland (grant no. 325022 and 352968), the Kone Foundation, and the Flagship programme "ACCC" of the Research Council of Finland (grant nos. 337550 and 337551).





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

660 1