# Peer review of "Accounting for effects of coagulation and model uncertainties in particle number concentration estimates based on measurements from sampling lines – A Bayesian inversion approach with SLIC v1.0"

_EGUsphere, 2024_

## Author Response (AR1)

**Response to reviewers**

Matti Niskanen

University of Eastern Finland

February 12, 2025

**Title: Accounting for effects of coagulation and model uncertainties in particle number concentration estimates based on measurements from sampling lines – A Bayesian inversion approach with SLIC v1.0**

Dear reviewers,

We thank you all for the feedback and kind words on the paper. In this document we have replied to the specific questions/comments raised by reviewers 1 and 2. The reviewer comments will be **referenced in bold font**.

Kind regards, Matti Niskanen

**Reviewer 1**

**I have written down some comments that the authors can choose to address in a possible revision, but none of these comments is blocking (i.e., feel free to ignore them).**

1. **In Section 2, I understand (and support) the decision by the authors to only address select physics processes that are most relevant here. For completion, there are other processes that can be accounted for in potential expansion of the model. Below, I give one example about charges. I encourage the authors to include more examples of potential processes that could be accounted for — are there processes you suspect could be important for future studies?**

   Author's response: The type of aerosol and coagulation processes that need to be modelled for reliable inference depend naturally on the specific experiment. As the reviewer also points out, the inversion framework is readily extendable to account for other processes such as Coulombic interactions in coagulation, but also different kinds of aerosol processes such as condensation and evaporation. We have added references to the paper on the effects of electric charge and turbulent flow on coagulation, see lines 116–119.

2. **In Section 4, it is readily clear how one would assess the synthetic data (we know the full ground truth) but how could one address the case of measurements in Section 4.3 where we don't know the full ground truth? Any thoughts on a potential \*experiment\* to further constrain the estimates (and test your method)?**

Author's response: Assessing the estimation error in the case of real measurements is of course difficult, since the ground truth isn't known. Unless we could somehow generate the emitted PSD exactly according to some specification, there's always going to be uncertainty on what the truth is. And in that case, quantifying the uncertainty is the best we can do, which is what we attempt in this paper as well.

A way to reduce uncertainty in the PSD estimates would be to make the sampling line as short as possible so that the coagulation effects are minimized, and in theory we could measure "at the source". In practice, though, there is always some minimum distance of sampling line needed, for example because the auxiliary heater was located outside while the measurement device was inside. An experiment following this idea was carried out by Liu *et al.* (10.1016/j.fuel.2021.121340), where they measured the effect of coagulation along an exhaust pipe in a laboratory setting. We cite this work in the introduction, see lines 43–47.

**Some comments I wrote while reading the manuscript:**

- **L105: provide a proper citation for SciPy (e.g., see this file https://github.com/scipy/scipy/blob/main/CITATION.bib)**

  Author's response: We added the suggested citation to the manuscript.

- **L115 and thereabouts: it is ok to ignore other factors, but a reader may be interested to follow up. I would add a few citations here to other works that dealt with other aspects of coagulation. For example, some recent work dealt with charging effects on collisions (e.g., 10.5194/acp-21-3827-2021, 10.5194/acp-23-6703-2023, 10.1029/2021GL092758, 10.1073/pnas.2313897121). You can probably find more references on other topics. There's no need to delve too deeply into any of this, but providing some references for the interested reader will be sufficient and helpful**

  Author's response: We've added some references to charging effects and turbulence effects on coagulation. See lines 116–119 in the paper.

- **L200 and thereabouts: Do you think charges on the walls play a role in deposition? See the following which is related to the above papers: 10.1080/02786826.2020.1757032**

  Author's response: It is possible that the charges on the walls increase the deposition rate, especially since exhaust particles are often electrically charged (according to 10.1016/j.jaerosci.2005.08.003). We don't, however, have specific data on auxiliary heater exhaust, whose combustion process can be quite different from the one of motor vehicle engines. We have now mentioned other mechanisms of wall deposition, including the effect of charges on the walls, with some citations in the theory section (see lines 204–208). Further, we mention electrostatic deposition as a potential major mechanism which we haven't modelled in the discussion (see lines 518 and 526–529).

- **L216: What's the n and B for? Please define them here (also, define the M below in L220 as well)**

Author's response: We've now clarified in the paper that the symbols refer to the lengths of the corresponding vectors. $n_B$ stands for number of bins and $n_M$ for number of measurement channels.

- **Section 2: I am not sure a full derivation of already established material (2.1.1, 2.1.2, 2.2) is needed (consider trimming). It's okay to keep as-is though. Sometimes repeating already known material is beneficial :)**

Author's response: It is true that most of Section 2 presents already established material. However, it does combine approaches from a few different publications with mutually different notations, which is why we felt that it is still worth it to present the models used in this paper in one place for ease of reference and consistent notation.

- **L375: "inversion crime" is vague; please elaborate**

Author's response: We have now added a definition of the inverse crime, see lines 382–386.

- **Fig 7: I would consider flipping the b panel (so that dn/dlogdp can still be on the y-axis). Also, why do the axes change? dN/dlogdm in an and then dN/dlogdp in b. Maybe a typo? Also, is there a normalized version of the density such that it adds up to 1?**

Author's response: Flipping the b-panel is something we also considered, but ultimately decided to keep it so that dN/dlogdm is on the x-axis, for the simple reason that we found it more readable when the plot follows the convention of plotting the independent variable (here, dN/dlogdm) on the horizontal axis and the dependent one (density) on the vertical axis. We hope that the remark on the axis in the text clears up the potential confusion for readers. We have fixed the typo, so changed dp -> $d_m$. Finally, the densities we have plotted do sum to one when integrated over their support.

- **L466: Why not drop the zeros for these plots? (i.e., make them NaN)**

Author's response: We set these values to NaN as suggested. This affects Figures 10, 11, and 12.

- **L500 and thereabouts: see my above comment about charging; in general, I think you can cite some works on extra confounding factors, but no need to delve too deep into them. As you cite the Seinfeld & Pandis book on coagulation, see the appendix of the chapter where coagulation is treated for more info (Appendix 13.1 in the 2016 book, 3rd edition)**

Author's response: Also referring to our comment above, we have briefly discussed effects of Coulombic forces and turbulence by providing some references in the theory section (see lines 116–119), and stated that these effects could be included in the model if there is a need to. To avoid repetition, we chose to not mention that in the discussion any more.

- **L508: there might be additional wall losses too (related to charges on walls, smoothness of walls, wetness of walls, etc.)**

Author's response: It is definitely possible that these factors have an effect on the wall deposition in the sampling ling. However, predicting them or even roughly estimating the magnitude of their effect is more difficult than that of for example diffusional or intertial impaction losses (stated for example in *Aerosol Technology* by Hinds (1999)). We have now

mentioned additional mechanisms of wall deposition both in the theory section (see lines 204–208) and the discussion (see lines 518 and 526–529).

**Reviewer 2**

**In section 2.2 there is discussion regarding wall deposition. It is stated that aerosol particles adhere when they collide with a surface, does their adherence change the deposition rate for future aerosols? I.e. does deposition/diffusion accelerate or decelerate any future deposition?**

Author's response: The depositing aerosol particles may form an uneven surface on the inside of the sampling line which may in turn cause an increase in the deposition rate. This can, in principle, be taken into account using wall roughness correlations based on experiments (e.g. Shimada et al., 1987), but has not been considered here. We have now (also in response to Reviewer 1) mentioned and cited some additional mechanisms of wall deposition, including effects of turbulence. See lines 204–208, 518, and 526–529.

**And one more complicated question: Could this method be applied to sampling other atmospheric pollutants?**

Author's response: Yes absolutely, the method is applicable to any measurement where coagulation is suspected to distort the size distribution. The method can also be extended to account for other aerosol processes in addition to coagulation (such as condensation/evaporation), if these are present in the measurement process. We have now mentioned this in the conclusion, see lines 535–536.